# Mind the Gap: Mitigating the Distribution Gap in Graph Few-shot Learning

**Chunhui Zhang**                                                        *chunhuizhang@brandeis.edu*
*Brandeis University, MA, USA*

**Hongfu Liu**                                                              *hongfuliu@brandeis.edu*
*Brandeis University, MA, USA*

**Jundong Li**                                                             *jundong@virginia.edu*
*University of Virginia, VA, USA*

**Yanfang Ye**                                                                *yye7@nd.edu*
*University of Notre Dame, IN, USA*

**Chuxu Zhang**                                                           *chuxuzhang@brandeis.edu*
*Brandeis University, MA, USA*

**Reviewed on OpenReview:** *https://openreview.net/forum?id=LEVbhNrLEL*

## Abstract

Prevailing supervised deep graph learning models often suffer from the issue of label scarcity, leading to performance degradation in the face of limited annotated data. Although numerous graph few-shot learning (GFL) methods have been developed to mitigate this problem, they tend to rely excessively on labeled data. This over-reliance on labeled data can result in impaired generalization ability in the test phase due to the existence of a *distribution gap*. Moreover, existing GFL methods lack a general purpose as their designs are coupled with task or data-specific characteristics. To address these shortcomings, we propose a novel **S**elf-**D**istilled **G**raph **F**ew-shot **L**earning framework (SDGFL) that is both general and effective. SDGFL leverages a self-distilled contrastive learning procedure to boost GFL. Specifically, our model first pre-trains a graph encoder with contrastive learning using unlabeled data. Later, the trained encoder is frozen as a teacher model to distill a student model with a contrastive loss. The distilled model is then fed to GFL. By learning data representation in a self-supervised manner, SDGFL effectively mitigates the distribution gap and enhances generalization ability. Furthermore, our proposed framework is task and data-independent, making it a versatile tool for general graph mining purposes. To evaluate the effectiveness of our proposed framework, we introduce an information-based measurement that quantifies its capability. Through comprehensive experiments, we demonstrate that SDGFL outperforms state-of-the-art baselines on various graph mining tasks across multiple datasets in the few-shot scenario. We also provide a quantitative measurement of SDGFL's superior performance in comparison to existing methods.

## 1 Introduction

Deep graph learning, such as graph neural networks (GNNs), has garnered significant attention for its exceptional performance in various domains, such as information systems (Kipf & Welling, 2017; Hamilton et al., 2017; Zhang et al., 2022a; Liu et al., 2023), molecular chemistry/biology (Jin et al., 2017; Hao et al., 2020), and recommendation (Ying et al., 2018; Fan et al., 2019). However, the success of GNNs typically requires a substantial amount of annotated data, which can be prohibitively expensive to obtain. To address

this challenge, graph few-shot learning (GFL) (Zhang et al., 2022b) has emerged as an area of active research aimed at improving performance in the face of limited labeled data.

Previous GFL models have been developed using meta-learning (or few-shot learning) techniques, such as metric-based approaches (Vinyals et al., 2016; Snell et al., 2017) or optimization-based algorithms (Finn et al., 2017). These models aim to quickly learn an effective GNN for new tasks with only a few labeled samples. GFL has been applied to a variety of graph mining tasks, including node classification (Zhou et al., 2019; Huang & Zitnik, 2020), relation prediction (Xiong et al., 2018; Lv et al., 2019; Zhang et al., 2020a), and graph classification (Chauhan et al., 2020; Ma et al., 2020). Despite considerable progress, most existing GFL models suffer from the following limitations: *(i) Impaired generalization.* Current GFL methods often rely too heavily on labeled data, which can result in limited generalization and transferability to new tasks due to the existence of a distribution shift between non-overlapping meta-training and meta-testing data. Without supervision signals from ground-truth labels, GFL may not learn an effective GNN for novel classes of test data, thus limiting its overall performance. *(ii) Constrained design.* Most current GFL methods lack a general purpose, as they assume that the designated task is universally the same prior across different graph tasks or datasets, which is not always guaranteed. For instance, GSM (Chauhan et al., 2020) requires the manual definition of a superclass of graphs, which cannot be extended to node-level tasks. The task or data-specific design limits the applicability of GFL to different graph mining tasks.

The challenges highlighted earlier underscore the need for a novel, generic GFL framework that can learn a transferable, effective, and generalizable GNN for various graph mining tasks with limited labeled data. Fortunately, contrastive learning has emerged as a promising approach to reduce the dependence on labeled data and learn label-independent yet transferable representations from unsupervised pretext tasks for vision, language, and graphs (Chen et al., 2020; Gao et al., 2021; You et al., 2020; Sohn et al., 2020). Consequently, we propose a novel **S**elf-**D**istilled **G**raph **F**ew-shot **L**earning framework (SDGFL) that leverages contrastive learning to enhance GFL.

Specifically, our proposed framework pre-trains a GNN by minimizing the contrastive loss between the embeddings of two views generated from two augmented graphs. Then, in order to learn more general and transferable representations for fast adaptation in GFL, we introduce a self-distillation step, an implicit ensemble of two pre-trained models (teacher and student), which reduces the variance (Allen-Zhu & Li, 2023) to improve the pre-training: the pre-trained GNN is frozen as a teacher model and utilized in the contrastive framework to distill a randomly initialized student model by minimizing the agreement of the embeddings generated by the two models. Both pre-training and the distillation steps can operate without requiring labeled data before the meta-learning phase. Finally, the distilled student model is taken as the initialized model for GFL in few-shot graph mining tasks. SDGFL learns graph representation in a self-supervised manner, effectively mitigating the negative impact of distribution shift, while producing transferable and discriminative graph representation for new tasks in the test data. Furthermore, our simple and generic framework is applicable to different graph mining tasks. To quantitatively measure the capability of SDGFL, we introduce an information-based method that measures the quality of learned node (or graph) embeddings on each layer of the model. Specifically, we assign each node a learnable variable as a noise and train these variables to maximize the entropy while keeping the change of output as small as possible.

To summarize, our contributions in this work are:

- We develop a general and effective SDGFL framework that leverages a self-distilled contrastive learning procedure to enhance GFL. SDGFL mitigates the impact of distribution shift and has a task and data-independent capacity for general graph mining purposes.

- We introduce an information-based method to quantitatively measure the capability of SDGFL by evaluating the quality of learned node (or graph) embeddings. To our knowledge, this is the first study that explores GFL model measurement.

- Comprehensive experiments on multiple graph datasets demonstrate that SDGFL outperforms state-of-the-art methods for both node classification and graph classification tasks in the few-shot scenario. Additional measurement results further confirm that SDGFL learns better node (or graph) embeddings than baseline methods.

## 2 Related Work

**Few-Shot Learning on Graphs**. In recent years, several GFL models have been proposed to solve various graph mining problems in the face of label sparsity, such as node classification (Yao et al., 2020; Ding et al., 2020; Huang & Zitnik, 2020; Wang et al., 2022; Zhang et al., 2022b;c), relation prediction (Xiong et al., 2018; Lv et al., 2019; Chen et al., 2019; Zhang et al., 2020a;b; 2022d), and graph classification (Chauhan et al., 2020; Ma et al., 2020; Guo et al., 2021; Wang et al., 2021; Zhang et al., 2023b). These models are built on meta-learning (or few-shot learning) techniques that can be categorized into two major groups: (1) metric-based approaches, which learn effective similarity metrics between few-shot support data and query data (Vinyals et al., 2016; Snell et al., 2017); (2) optimization-based algorithms (Finn et al., 2017), which aim to learn well-initialized GNN parameters that can be quickly adapted to new graph tasks with few labeled data. For example, GPN (Ding et al., 2020) conducts node informativeness propagation to build weighted class prototypes for a distance-based node classifier. The second group proposes to learn well-initialized GNN parameters that can be fast adapted to new graph tasks with few labeled data. For instance, G-Meta (Huang & Zitnik, 2020) builds local subgraphs to extract subgraph-specific information and optimizes GNN via MAML (Finn et al., 2017). While previous efforts have relied on labeled data and had task and data-specific designs, we aim to develop a novel framework that explores unlabeled data and has a generic design for general graph mining purposes.

**Self-Supervised Learning on Graphs**. Self-supervised graph learning (SGL) has recently received considerable attention due to its effectiveness in pre-training GNNs and its competitive performance in various graph mining applications. Previous SGL models can be categorized into two main groups: generative learning and contrastive learning (Liu et al., 2020; Sohn et al., 2020; Zhao et al., 2021; Yu et al., 2022; Yue et al., 2022; Qian et al., 2022; Zhang et al., 2023a; Tian et al., 2023). The generative models learn the graph representation by recovering feature or structural information on the graph. The task can recover only the adjacency matrix alone (You et al., 2018) or together with the node features (Hu et al., 2020b). As for the contrastive methods, they first define the node context, which can be node-level or graph-level instances. Then, they perform contrastive learning either by maximizing the mutual information between the node-context pairs (Hassani & Ahmadi, 2020; Velickovic et al., 2019; Sun et al., 2020) or by discriminating the context instances (Qiu et al., 2020; Zhu et al., 2021). In addition to the above strategy, random propagation has recently applied graph augmentation (Rong et al., 2020) for semi-supervised learning (Feng et al., 2020). Motivated by the success of SGL, we propose to use it to improve GFL.

## 3 Preliminary

**GNNs**. A graph is represented as $G = (V, E, X)$, where $V$ is the set of nodes, $E \subseteq V \times V$ is the set of edges, and $X$ is the set of node attributes. GNNs (Hamilton et al., 2017; Xu et al., 2019; Zhang et al., 2019) learn compact representations (embeddings) by considering both graph structure $E$ and node attribute $X$. To be specific, let $f_\theta(\cdot)$ denote a GNN encoder with parameter $\theta$, the updated embedding of node $v$ at the $l$-th layer of GNN can be formulated as:

$$h_v^{(l)} = \mathcal{M}(h_v^{(l-1)}, \{h_u^{(l-1)} \mid \forall u \in \mathcal{N}_v\}; \theta),\tag{1}$$

where $\mathcal{N}_v$ denotes the neighbor set of $v$; $\mathcal{M}(\cdot)$ is the message passing function for neighbor information aggregation, such as a mean pooling layer followed by a fully-connected (FC) layer; $h_v^{(0)}$ is initialized with node attribute $X_v$. The whole graph embedding can be computed over all nodes' embeddings as:

$$h_G^{(l)} = \text{READOUT}\{h_v^{(l)} \mid \forall v \in V\},\tag{2}$$

where the READOUT function can be a simple permutation invariant function such as summation.

**GFL Setting and Problem**. Let $\mathcal{C}_{base}$ and $\mathcal{C}_{novel}$ denote the base classes set and novel (new) classes set in training data $\mathcal{T}_{train}$ and testing data $\mathcal{T}_{test}$, respectively. Similar to the general meta-learning problem (Finn et al., 2017), the graph few-shot learning (GFL) aims to train a GNN encoder $f_\theta(\cdot)$ over $\mathcal{C}_{base}$ that can be quickly adapted to $\mathcal{C}_{novel}$ with limited labeled data per class. The base and novel classes are disjoint, denoted

as $\mathcal{C}_{base} \cap \mathcal{C}_{novel} = \varnothing$. In $K$-shot setting, a batch of classes (tasks) is randomly sampled from $\mathcal{C}_{base}$ during the meta-training phase, where $K$ labeled instances are used to form the support set $\mathcal{S}$ for model training and the remaining instances are taken as the query set $\mathcal{Q}$ for model evaluation. After sufficient training, the model is further transferred to the meta-testing phase to conduct $N$-way classification over $\mathcal{C}_{novel}$ ($N$ is the number of novel classes), where each class is only with $K$ labeled instances. The GFL framework can be applied to various graph mining problems, such as node classification or graph classification, depending on the class definition. In this study, we consider both node classification and graph classification problems under the few-shot setting, formally defined as follows:

**Problem 1** *Few-Shot Node Classification. Given a graph $G = (V, E, X)$ and labeled nodes of $\mathcal{C}_{base}$, the problem is to learn a GNN $f_\theta(\cdot)$ to classify nodes of $\mathcal{C}_{novel}$, where each class in $\mathcal{C}_{novel}$ only has few labeled nodes.*

**Problem 2** *Few-Shot Graph Classification. Given a set of graphs $\mathcal{G}$ and labeled graphs of $\mathcal{C}_{base}$, the problem is to learn a GNN $f_\theta(\cdot)$ to classify graphs of $\mathcal{C}_{novel}$, where each class in $\mathcal{C}_{novel}$ only has few labeled graphs.*

In contrast to previous studies that rely solely on labeled data of $\mathcal{T}_{train}$ and $\mathcal{T}_{test}$ for GFL model training and adaptation, our approach leverages both labeled data and unlabeled graph information to learn a GFL model for solving the above problems.

## 4 Methodology

Figure 1 depicts the proposed SDGCL framework, which comprises two main phases: self-distilled graph contrastive learning and graph few-shot learning (GFL). The first phase (Figure 1(a)) involves pre-training a GNN encoder with contrastive learning, followed by knowledge distillation to enhance the pre-trained GNN in a self-supervised manner. The distilled GNN is subsequently fed into the GFL phase (Figure 1(b)) for few-shot graph mining tasks. Additionally, we introduce an information-based approach to quantify the superiority of SDGCL.

### 4.1 Self-Distilled Graph Contrastive Learning

**GNN Contrastive Pre-training**. In the first phase, we employ contrastive learning to pre-train the GNN. Inspired by the representation bootstrapping technique (Grill et al., 2020), our method learns node (or graph) representations by discriminating context instances. We introduce two GNN encoders: an online GNN $f_\theta(\cdot)$ and a target GNN $f_\xi(\cdot)$, to encode two randomly augmented views of a given graph. The online GNN is supervised under the target GNN's output, while the target GNN is updated by the online GNN's exponential moving average. Figure 1(a) shows the contrastive pre-training step.

*Graph Augmentation:* We process the given graph $G$ with random data augmentations to generate a contrastive pair $(G', G'')$ as input for the two GNN branches (online branch and target branch) used in GNN training. We apply a combination of stochastic node feature masking, edge removal, and node dropping with constant probabilities for graph augmentation.

*GNN Update:* Using the generated graph pair $(G', G'')$, the online GNN $f_\theta(\cdot)$ and the target GNN $f_\xi(\cdot)$ are respectively utilized to process $G'$ and $G''$ for node (or graph) embedding generation. Both GNNs have the same architecture, while a two-layer FC (one-layer FC) is attached after an online GNN (target GNN) to refine embedding. To prevent the prediction of the online model from being exactly the same as the output of the target model and avoid the learned representation collapse, two branches have different FC layers. To enforce the online GNN's embeddings $z_\theta$ to approximate the target GNN's embeddings $h_\xi$, we formulate the mean squared error between them as the objective function:

$$\mathcal{L}_{\theta,\xi} = \|z_\theta - h_\xi\|_2^2 = 2 - 2 \cdot \frac{z_\theta, h_\xi}{\|z_\theta\|_2 \cdot \|h_\xi\|_2}. \tag{3}$$

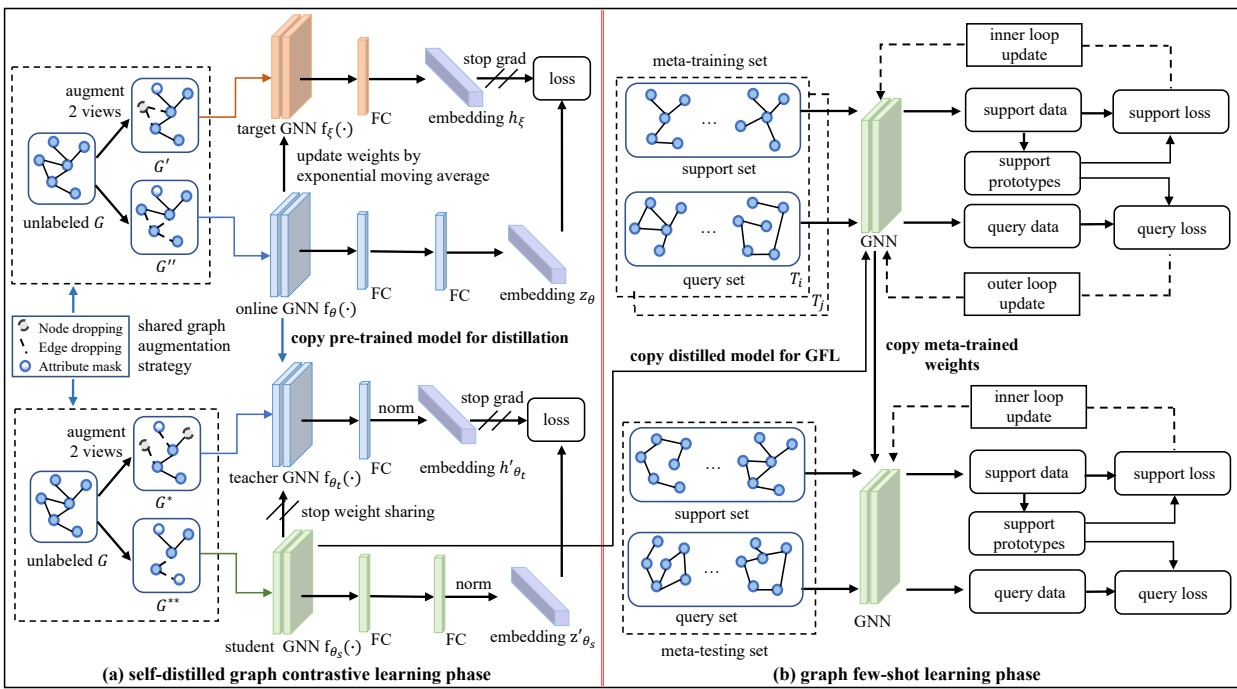

Figure 1: The SDGCL framework consists of two phases: (a) self-distilled graph contrastive learning, where a GNN encoder is pre-trained with stop-grad contrastive learning (Grill et al., 2020; Chen & He, 2021) and further evaluated with knowledge distillation in a self-supervised manner; (b) graph few-shot learning, which takes the distilled student network as the initialized model and uses a meta-learning algorithm for model optimization.

We update the parameters $\theta$ of the online GNN using the Adam optimizer (Kingma & Ba, 2015):

$$\theta \leftarrow \text{Adam}(\theta, \nabla_\theta \mathcal{L}_{\theta,\xi}, \eta), \tag{4}$$

where $\eta$ is the learning rate. We use the target GNN to provide the regression target to supervise the online GNN, and its parameters $\xi$ are updated as the exponential moving average (EMA) of the online GNN parameters $\theta$. More precisely, $\xi$ is updated as follows:

$$\xi \leftarrow \tau\xi + (1-\tau)\theta, \tag{5}$$

where $\tau \in [0,1]$ is the decay rate. Note that the target GNN stops the backpropagation from $\mathcal{L}_{\theta,\xi}$, and it is only updated by EMA.

**Contrastive Distillation**. With the pre-trained GNN $f_\theta(\cdot)$ obtained in the previous step, we propose a method called Contrastive Distillation to enhance the performance of a pre-trained GNN $f_\theta(\cdot)$. Our approach is inspired by the Born-again strategy (Furlanello et al., 2018), which posits that a well-trained teacher can improve a randomly initialized identical student. The distillation step adopts a similar contrastive framework as the previous step, as shown in Figure 1(a). Specifically, we first use the pre-trained GNN $f_\theta(\cdot)$ as the teacher model $f_{\theta_t}(\cdot)$, and then generate two augmented views ($G^*$, $G^{**}$) of a graph $G$ to be fed into both the teacher and the student models ($f_{\theta_t}(\cdot)$ and $f_{\theta_s}(\cdot)$), respectively. The teacher model is then frozen and used to distill the student model $f_{\theta_s}(\cdot)$. To ensure that the student model approximates the teacher model, we use a contrastive framework, as illustrated in Figure 1 (a). Specifically, we force the student's normalized output to approximate the teacher's normalized output, which is represented as follows:

$$\mathcal{L}_{\theta_s} = \|z'_{\theta_s} - h'_{\theta_t}\|_2^2 = 2 - 2 \cdot \frac{z'_{\theta_s}, h'_{\theta_t}}{\|z'_{\theta_s}\|_2 \cdot \|h'_{\theta_t}\|_2}, \tag{6}$$

$$z'_{\theta_s} = \frac{z_{\theta_s}}{\|z_{\theta_s}\|_2}, \quad h'_{\theta_t} = \frac{h_{\theta_t}}{\|h_{\theta_t}\|_2}, \tag{7}$$

where $z_{\theta_s}$ and $h_{\theta_t}$ are the output embeddings of the student and teacher models, respectively. The student model is then updated using the Adam optimizer, as follows:

$$\theta_s \leftarrow \text{Adam}(\theta_s, \nabla_{\theta_s}\mathcal{L}_{\theta_s}, \eta). \tag{8}$$

In contrast to the target GNN update in contrastive pre-training that uses the EMA method (Eqn. 5), the teacher model in our method is frozen and can be seen as a special case of EMA, which is represented as follows:

$$\theta_t \leftarrow \tau\theta_t + (1 - \tau)\theta_s, \quad \tau = 1. \tag{9}$$

## 4.2 Graph Few-Shot Learning

In the GFL phase, we take the distilled student GNN $f_{\theta_s}(\cdot)$ generated in the previous phase as the initialized GNN model using an optimization-based algorithm called model-agnostic meta-learning (MAML) (Finn et al., 2017). During meta-training, we compute the task-specific parameters $\theta'_{s,i}$ for task $\mathcal{T}_i$ using a number of gradient descent updates over the support set $\mathcal{S}_i$ of $\mathcal{T}_i$ (i.e., inner-loop), as follows:

$$\theta'_{s,i} \leftarrow \theta_s - \alpha\nabla_{\theta_s}\mathcal{L}^{\mathcal{S}_i}_{\mathcal{T}_i}(f_{\theta_s}), \tag{10}$$

where $\alpha$ is the learning step size, and $\mathcal{L}^{\mathcal{S}_i}_{\mathcal{T}_i}$ denotes the downstream task loss over $\mathcal{S}_i$. To perform node or graph classification, we use the prototypical loss (Snell et al., 2017), which utilizes embeddings extracted from the support set by a neural network as the class prototype, and the query set is classified according to the distance between its embeddings and prototypes. We then utilize the task-specific parameter $\theta'_{s,i}$ to compute the loss over the query set $\mathcal{Q}_i$ of $\mathcal{T}_i$, which is denoted as $\mathcal{L}^{\mathcal{Q}_i}_{\mathcal{T}_i}(f_{\theta'_{s,i}})$. The losses of a batch of randomly sampled tasks are then summed up to update the model parameters $\theta_s$ (i.e., outer-loop), as follows:

$$\theta_s \leftarrow \theta_s - \beta\nabla_{\theta_s}\sum_i\mathcal{L}^{\mathcal{Q}_i}_{\mathcal{T}_i}(f_{\theta'_{s,i}}), \tag{11}$$

where $\beta$ is the learning step size. During meta-testing, we apply the same procedure, but using the final meta-updated parameter $\theta_s$ for novel tasks, without the outer-loop. $\theta_s$ is learned from knowledge across meta-training tasks and represents the optimal parameter for quickly adapting to novel tasks. It is important to note that the GFL algorithm can be applied to various graph mining problems by changing the meaning of each task. Specifically, for node classification (or graph classification), each task corresponds to a node class (or graph class).

## 4.3 Quantitative Measurement of GFL

The previous GFL studies target developing better methods in performance while none of them has thought about model capability measurement. To fill the gap, we extend the neural network model capability measurement method proposed by Ma et al. (Ma et al., 2019) to graph data and use mutual information ($MI$) to measure the information of the input graph $G$ encoded by the hidden state $Z$ of a GFL model $f$. Specifically, the mutual information $MI(G; Z)$ is given by:

$$MI(G; Z) = H(G) - H(G|Z), \tag{12}$$

where $H(\cdot)$ denotes the entropy, $H(G)$ is a constant, and $H(G|Z)$ represents the amount of discarded information after $G$ is processed by $f$ and encoded by $Z$. We can compute $H(G|Z)$ by decomposing it into the node level as follows:

$$H(G|Z) = \int_{\mathbf{z} \in Z} p(\mathbf{z})H(G|\mathbf{z})d\mathbf{z}, \tag{13}$$

where $\mathbf{z} = f(x)$ denotes the hidden state corresponding to attribute $x$ of a node. To disentangle information components of individual nodes from the whole graph, we assume that each node is independent of the others and have:

$$H(G|\mathbf{z}) = \sum_i H(x_i|\mathbf{z}), \tag{14}$$

where $x_i$ denotes a random variable of the $i$-th node attribute in the graph. Then, we introduce a noise perturbation-based method to approximate $H(x_i|\mathbf{z})$. Specifically, let $\widetilde{x}_i = x_i + \epsilon_i$ $(\epsilon_i \sim \mathcal{N}(0, \mathbf{\Sigma}_i = \sigma_i^2 \mathbf{I}))$ and we aim to optimize the following loss function:

$$\mathcal{L}(\sigma) = \mathbb{E}_\epsilon \|\mathbf{f}(\mathbf{x_i}) - \mathbf{z}\|^2 - \beta \sum_{\mathbf{i=1}}^{\mathbf{n}} \mathbf{H}(\mathbf{x_i}|\mathbf{z})|_{\epsilon_\mathbf{i} \sim \mathcal{N}(\mathbf{0}, \sigma_\mathbf{i}^2 \mathbf{I})}, \tag{15}$$

where $\sigma = [\sigma_1, \sigma_2, \cdots, \sigma_n]$ are learnable parameters and $\beta$ is a trade-off weight. In particular, The first term of the above objective minimizes the difference between the encoded embedding of noisy input and the hidden state, while the second term encourages a high conditional entropy according to the maximum entropy principle. In this way, we have $p(\widetilde{x}_i|\mathbf{z}) = p(\epsilon_i)$ and $H(x_i|\mathbf{z})$ is approximated by $H(\widetilde{x}_i|\mathbf{z})$ as follows:

$$H(\widetilde{x}_i|\mathbf{z}) = p(\widetilde{x}_i|\mathbf{z}) \log p(\widetilde{x}_i|\mathbf{z}) \propto \log \sigma_i + C, \tag{16}$$

where $C = \frac{1}{2}\log(2\pi e)$. By taking Eqn. 16 into Eqn. 15, we can optimize $\mathcal{L}(\sigma)$ using the Adam optimizer to obtain the optimal $\sigma$ for computing the overall discarded information $H(G|Z)$. Comparing the discarded information of different GFL models allows us to measure their capabilities. Ideally, we want the GFL model to encode valid node (or graph) embeddings as much as possible, which means discarding as little information as possible.

# 5 Experiments

We have performed comprehensive experiments on multiple graph datasets to evaluate our model's performance against state-of-the-art models. In this section, we first describe our experimental settings and then present our findings by comparing the performance of various models. Finally, we compute the discarded information to demonstrate the capabilities of different GFL models. Additional experimental results can be found in Appendix C.

## 5.1 Experimental Setup

**Datasets**. To conduct our experiments, we used multiple graph datasets. Specifically, for the node classification task, we used ogbn-arxiv (Hu et al., 2020a), Tissue-PPI (Hamilton et al., 2017), Fold-PPI (Zitnik & Leskovec, 2017), Cora (Sen et al., 2008), and Citeseer (Sen et al., 2008). For the graph classification task, we used the datasets in (Chauhan et al., 2020), namely, Letter-High, Triangles, Reddit-12K, and Enzymes. Appendix A provides more detailed information about the datasets.

**Baseline Methods**. We employ a diverse range of baseline methods for model comparison in the two tasks. For few-shot node classification, we use node2vec (Grover & Leskovec, 2016), DeepWalk (Perozzi et al., 2014), Meta-GNN (Zhou et al., 2019), FS-GIN (Xu et al., 2019), FS-SGC (Wu et al., 2019), No-Finetune (Triantafillou et al., 2019), Finetune (Triantafillou et al., 2019), KNN (Triantafillou et al., 2019), ProtoNet (Snell et al., 2017), MAML (Finn et al., 2017), G-Meta (Huang & Zitnik, 2020), and TENT (Wang et al., 2022). For few-shot graph classification, we utilize WL (Shervashidze et al., 2011), Graphlet (Shervashidze et al., 2009), AWE (Ivanov & Burnaev, 2018), Graph2Vec (Narayanan et al., 2017), Diffpool (Lee et al., 2019), CapsGNN (Xinyi & Chen, 2018), GIN (Xu et al., 2019), GIN-KNN (Xu et al., 2019), GSM-GCN (Chauhan et al., 2020), GSM-GAT (Chauhan et al., 2020), and AS-MAML (Ma et al., 2020). Details of the baseline methods are presented in Appendix B.

**Experimental Settings**. In our proposed SDGCL approach, we adopt GCN (Kipf & Welling, 2017) as the GNN backbone for the node classification task, which includes a two-layer graph convolution and a one-layer fully connected layer. For the graph classification task, we add an additional average pooling operation as a readout layer. In the pretraining phase, we pretrain the GNN on the unlabeled graph dataset using contrastive learning. To augment graph data, we randomly drop 15% of the nodes, remove 15% of the edges, and mask 20% of the node features. The mini-batch size is set to 2,048, and we use a learning rate of 0.05 with a decay factor of 0.9. Furthermore, we set the $\tau$ value for exponential moving average to 0.999. We consider both inductive and transductive settings for our experiments. In the inductive setting (**SDGCL-I**),

Table 1: Few-shot node classification results. Baselines are designed for inductive setting only. The best results in our methods are highlighted in bold, while the best results in baselines are underlined. Also, baselines are implemented under inductive settings as stated in their original papers. -I and -T denote inductive and transductive settings of SDGCL, respectively. Teacher indicates the SDGCL model without knowledge distillation.

| Method | Tissue-PPI | | Fold-PPI | | Cora | | Citeseer | | ogbn-arxiv | |
|---|---|---|---|---|---|---|---|---|---|---|
| | 3-shot | 5-shot | 3-shot | 5-shot | 3-shot | 5-shot | 3-shot | 5-shot | 3-shot | 5-shot |
| node2vec | 48.5±3.3 | 49.3±3.9 | 36.6±3.7 | 37.4±1.9 | 25.7±1.3 | 26.9±3.0 | 20.0±2.5 | 21.7±2.9 | 28.9±4.0 | 29.5±3.7 |
| DeepWalk | 46.2±4.8 | 47.4±3.6 | 35.0±4.4 | 36.3±3.2 | 25.6±0.8 | 26.7±2.0 | 21.2±0.6 | 22.6±2.7 | 30.3±2.1 | 31.5±3.4 |
| Meta-GNN | 50.8±8.1 | 53.5±1.5 | 30.8±5.4 | 33.5±2.1 | 76.8±0.9 | 79.2±1.9 | 69.4±1.4 | 72.6±1.9 | 27.3±1.2 | 30.2±3.6 |
| FS-GIN | 49.2±2.4 | 51.5±3.0 | 36.7±2.1 | 39.1±1.4 | 53.5±1.6 | 56.2±2.8 | 50.2±2.6 | 53.2±3.8 | 33.6±4.2 | 36.8±2.5 |
| FS-SGC | 49.8±3.8 | 52.3±2.2 | 38.0±1.6 | 40.9±3.9 | 57.2±2.1 | 60.3±1.2 | 52.0±2.1 | 54.4±2.5 | 34.7±0.5 | 37.3±1.0 |
| No-Finetune | 51.6±0.6 | 55.0±2.1 | 37.6±1.7 | 39.9±3.6 | 61.2±1.2 | 64.5±1.3 | 54.9±1.7 | 58.3±2.5 | 36.4±1.4 | 38.8±2.0 |
| Finetune | 52.1±1.3 | 54.3±2.4 | 37.0±2.2 | 40.0±2.6 | 63.5±0.8 | 65.7±2.1 | 57.8±1.8 | 59.0±2.9 | 35.9±1.0 | 38.6±2.5 |
| KNN | 61.9±2.5 | 65.2±3.2 | 43.3±3.4 | 46.2±1.9 | 67.8±1.4 | 70.3±3.6 | 60.6±1.4 | 63.2±1.6 | 39.2±1.5 | 42.3±1.8 |
| ProtoNet | 54.6±2.5 | 57.5±2.9 | 38.2±3.1 | 41.3±1.1 | 42.6±3.7 | 56.6±2.9 | 55.5±1.5 | 58.0±3.7 | 37.2±1.7 | 39.7±1.7 |
| MAML | 74.5±5.1 | 77.4±2.7 | 48.2±6.2 | 51.3±3.3 | 65.7±0.9 | 68.8±1.1 | 63.1±1.6 | 65.7±1.7 | 38.9±2.1 | 41.3±2.4 |
| GPN | 77.3±3.0 | 79.0±3.6 | 57.0±4.7 | 58.2±3.7 | 73.1±2.2 | 76.1±2.2 | 68.3±1.4 | 71.1±2.0 | 44.4±3.5 | 48.2±4.0 |
| RALE | 76.6±3.3 | 79.2±3.3 | 57.8±4.5 | 58.8±3.3 | 62.8±3.1 | 65.9±3.2 | 69.9±2.3 | 71.3±2.2 | 45.1±2.7 | 47.8±1.5 |
| G-Meta | 76.8±2.9 | 79.4±2.6 | 56.1±5.9 | 59.0±2.5 | 71.9±2.9 | 74.5±2.0 | 67.8±2.2 | 70.8±3.8 | 45.1±3.2 | 48.2±3.1 |
| TENT | - | - | - | - | 64.8±4.1 | 69.2±4.5 | 54.2±3.4 | 62.0±2.3 | 55.6±3.1 | 62.9±3.7 |
| Teacher-I | 77.9±2.6 | 80.8±1.7 | 58.8±3.4 | 61.3±3.6 | 78.2±1.3 | 80.9±1.9 | 70.0±1.3 | 72.7±1.8 | 52.0±2.0 | 54.9±1.6 |
| SDGCL-I | 78.7±2.8 | 81.5±3.6 | 59.5±4.1 | 62.0±2.0 | 78.5±1.5 | 81.2±1.5 | 70.6±1.2 | 73.1±2.0 | 52.8±1.8 | 55.6±1.1 |
| Teacher-T | 79.8±3.1 | 82.9±1.3 | 63.0±3.6 | 65.6±2.2 | 80.0±2.7 | 82.6±1.9 | 72.1±1.1 | 75.0±1.5 | 54.3±2.7 | 58.4±3.9 |
| SDGCL-T | **80.9±3.0** | **84.1±3.2** | **66.9±3.4** | **69.0±2.7** | **80.7±1.9** | **83.5±3.0** | **72.6±1.6** | **75.3±2.0** | **55.2±2.5** | **58.7±2.7** |

we only use the unlabeled data in the training set, whereas in the transductive setting (**SDGCL-T**), we use the unlabeled data in both the training and testing sets. For pre-trained models without knowledge distillation, we denote them as **Teacher-I** and **Teacher-T** for the two settings, respectively. In the GFL phase, we utilize MAML for fine-tuning the model. Our SDGCL implementation is based on PyTorch, and we train it on NVIDIA V100 GPUs.

## 5.2 Few-Shot Node Classification

**Overall Performance**. Table 1 presents the performances of all models for 3/5-shot node classification. Our observations are as follows: 1) SDGCL outperforms all baseline methods on all datasets, demonstrating its superiority for few-shot node classification; (2) The improvement of SDGCL-I over baseline methods ranges from 1.1% to 50% (3-shot) and from 2.1% to 51% (5-shot), while the improvement of SDGCL-T ranges from 4.8% to 55.0% (3-shot) and from 4.5% to 56.6% (5-shot). These significant improvements demonstrate the effectiveness of contrastive pre-training and self-distillation in learning rich node representations from unlabeled graph data and addressing the label-hungry issue; (3) SDGCL-T (or Teacher-T) outperforms SDGCL-I (or Teacher-I), indicating that the model benefits from unlabeled data in the meta-testing set, thus improving generalization ability over testing data; (4) The contrastive distillation employed in SDGCL leads to an additional boost compared with the pre-trained teacher model without any label cost in distillation, demonstrating its effectiveness.

**Impact of Shot Number**. In Figure 2a, we present the performance of our model (SDGCL-T) under different shot numbers (1 to 5) compared to selected baselines. SDGCL *almost* outperforms other methods across all shot numbers, demonstrating the stability of our model for node classification. Note that we show results for two datasets only, while results for other datasets are presented in Appendix C.1.

**Impact of Training Label Rate**. To evaluate SDGCL's performance under different training label rates (10%, 20%, 30%, 40%, 50%, 100%), we compare it with baseline methods for 3-shot node classification, as shown in Figure 2b. Our model *nearly* consistently outperforms the baseline models across all label rates. Furthermore, the performance improvement of SDGCL over baseline models is more significant when the label rate becomes lower (e.g., 10%), demonstrating that the contrastive pre-training and self-distillation of SDGCL are effective in cases of label sparsity.

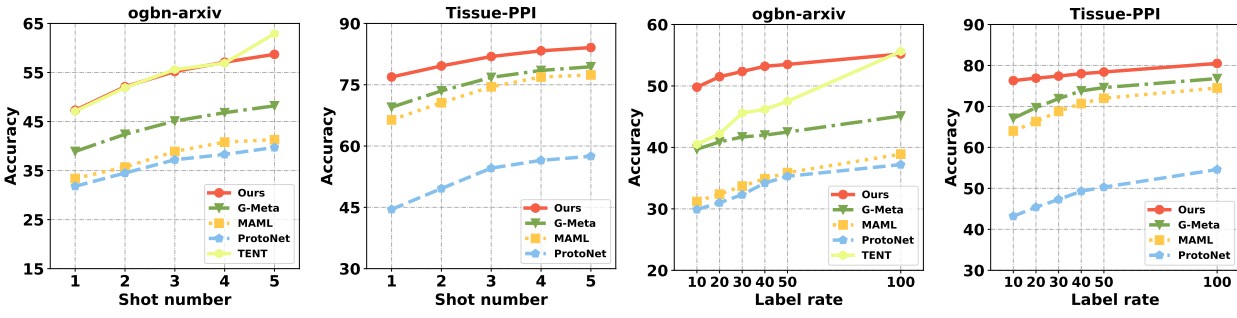

(a) Impact of shot number.    (b) Impact of label rate.

Figure 2: Impact of shot number and label rate on node classification.

**Impact of Data Augmentation**. Graph augmentation is a crucial step in contrastive learning of SDGCL, and it significantly affects model performance. We conduct experiments to evaluate the model performance with different augmentation strategies, including node dropping (**ND**), feature masking (**FM**), edge removing (**ER**), and their combinations. In Figure 3, we report the performances of SDGCL under these augmentation strategies. The combination of all three augmentation strategies outperforms single or two augmentations, demonstrating that various graph augmentations generate sufficient contrastive pairs to improve the model performance.

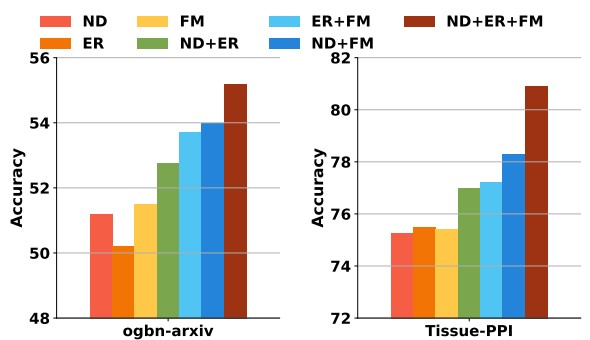

Figure 3: Impact of data aug. on node classification.

Table 2: Info. loss for node classification.

| Method | Layer | ogbn-arxiv | Tissue-PPI |
|--------|-------|-----------|-----------|
| G-Meta | GNN-1 | 385.15 | 788.30 |
| | GNN-2 | 383.39 | 789.78 |
| | FC | 374.42 | 753.85 |
| TENT | GNN-1 | 315.70 | - |
| | GNN-2 | 296.67 | - |
| | FC | 293.55 | - |
| SDGCL-I | GNN-1 | 305.29 | 663.38 |
| | GNN-2 | 300.29 | 650.59 |
| | FC | 290.99 | 612.98 |
| SDGCL-T | GNN-1 | **276.30** | **533.86** |
| | GNN-2 | **267.33** | **525.42** |
| | FC | **264.23** | **503.86** |

**Loss Information Comparison**. In Section 4.3, we propose to compute the amount of graph information discarded in the GFL model. Here we compare the results between SDGCL and a selected baseline method (G-Meta) in Table 2. It can be observed that the amount of discarded information in each layer of SDGCL is smaller than that of baseline models. This may be due to the fact that SDGCL can learn more label-independent information from unlabeled graph data, which demonstrates the superiority of SDGCL in learning node embeddings for node classification.

## 5.3 Few-Shot Graph Classification

**Overall Performance**. The presented results in Table 3 demonstrate the superiority of SDGCL over all baseline models in few-shot graph classification. Specifically, the improvement of SDGCL-I over baseline models ranges from 0.95% to 38.36% (5-shot) and from 0.8% to 34.33% (10-shot), while the improvement of SDGCL-T ranges from 5.65% to 46.21% (5-shot) and from 2.18% to 41.51% (10-shot). These results demonstrate the effectiveness of contrastive pre-training and self-distillation in learning informative graph embeddings from unlabeled data. Furthermore, the transductive setting of SDGCL-T outperforms the inductive setting of SDGCL-I, indicating the benefit of using unlabeled data in the testing set to improve generalization performance. Additionally, it is worth noting that the SDGCL model with contrastive

Table 3: Results of few-shot graph classification. Baselines are designed for inductive setting only. The best results in our methods are highlighted in bold, while the best results in baselines are underlined. Also, baselines are implemented under inductive settings as stated in their original papers. The suffixes -I and -T denote the inductive and transductive settings of SDGCL, respectively. The term "Teacher" refers to the SDGCL model without knowledge distillation.

| Method | Letter-High | | Triangles | | Reddit-12K | | Enzymes | |
|---|---|---|---|---|---|---|---|---|
| | 5-shot | 10-shot | 5-shot | 10-shot | 5-shot | 10-shot | 5-shot | 10-shot |
| WL | 65.27±7.67 | 68.39±4.69 | 51.25±4.02 | 53.26±2.95 | 40.26±5.17 | 42.57±3.69 | 55.78±4.72 | 58.47±3.84 |
| Graphlet | 33.76±6.94 | 37.59±4.60 | 40.17±3.18 | 43.76±3.09 | 33.76±6.94 | 37.59±4.60 | 53.17±5.92 | 55.30±3.78 |
| AWE | 40.60±3.91 | 42.20±2.87 | 39.36±3.85 | 42.58±3.11 | 30.24±2.34 | 33.44±2.04 | 43.75±1.85 | 45.58±2.11 |
| Graph2Vec | 66.12±5.21 | 68.17±4.26 | 48.38±3.85 | 50.16±4.15 | 27.85±4.21 | 29.97±3.17 | 55.88±4.86 | 58.22±4.30 |
| Diffpool | 58.69±6.39 | 61.59±5.21 | 64.17±5.87 | 67.12±4.29 | 35.24±5.69 | 37.43±3.94 | 45.64±4.56 | 49.64±4.23 |
| CapsGNN | 56.60±7.86 | 60.67±5.24 | 65.40±6.13 | 68.37±3.67 | 36.58±4.28 | 39.16±3.73 | 52.67±5.51 | 55.31±4.23 |
| GIN | 65.83±7.17 | 69.16±5.14 | 63.80±5.61 | 67.30±4.35 | 40.36±4.69 | 43.70±3.98 | 55.73±5.80 | 58.83±5.32 |
| GIN-KNN | 63.52±7.27 | 65.66±8.69 | 58.34±3.91 | 61.55±3.19 | 41.31±2.84 | 43.58±2.80 | 57.24±7.06 | 59.34±5.24 |
| GSM-GCN | 68.69±6.50 | 72.80±4.12 | 69.37±4.92 | 73.11±3.94 | 40.77±4.32 | 44.28±3.86 | 54.34±5.64 | 58.16±4.39 |
| GSM-GAT | 69.91±5.90 | 73.28±3.46 | 71.40±4.34 | 75.60±3.67 | 41.59±4.12 | 45.67±3.68 | 55.42±5.74 | 60.64±3.84 |
| AS-MAML | 70.23±1.53 | 73.19±1.17 | 71.56±1.17 | 75.56±2.39 | 41.90±1.65 | 45.66±1.11 | 56.03±1.85 | 60.79±2.74 |
| Teacher-I | 71.43±5.23 | 73.62±2.93 | 71.93±3.51 | 76.21±2.87 | 42.32±4.48 | 46.31±3.84 | 57.53±3.16 | 61.12±4.80 |
| SDGCL-I | 72.12±4.88 | 74.08±3.30 | 72.34±3.42 | 76.91±2.98 | 42.85±4.62 | 46.89±4.96 | 57.94±2.85 | 61.66±4.61 |
| Teacher-T | 75.20±4.34 | 78.35±1.95 | 78.55±3.75 | 81.03±3.37 | 44.80±4.85 | 48.95±4.03 | 59.85±2.34 | 62.30±3.29 |
| SDGCL-T | **75.97±5.02** | **79.10±4.23** | **79.32±4.05** | **81.78±3.30** | **45.55±3.67** | **49.32±4.21** | **60.34±4.04** | **62.97±2.92** |

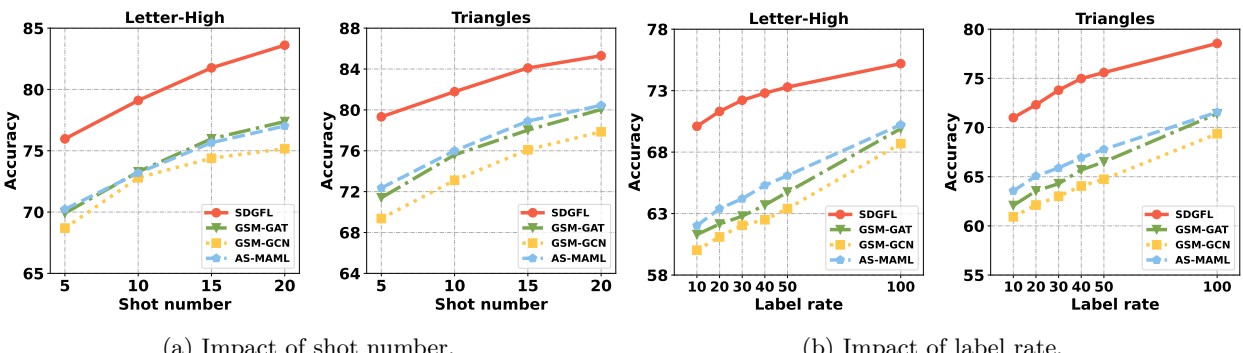

(a) Impact of shot number.  (b) Impact of label rate.

Figure 4: Impact of shot number and label rate on graph classification.

distillation (Teacher-T) outperforms the SDGCL model without distillation, highlighting the effectiveness of the proposed knowledge distillation method.

**Impact of Shot Number**. In Figure 4a, we present the results of our model (SDGCL) under varying shot numbers (5, 10, 15, 20) compared to select baseline methods. Similar to Figure 2a, SDGCL consistently outperforms the baseline models across varying shot numbers, demonstrating the robustness of SDGCL. We note that the results are presented for two datasets, namely Letter-High and Triangles, and the results for the other datasets are shown in Appendix C.2.

**Impact of Training Label Rate**. In Figure 4b, we show the performance of SDGCL under different training label rates in comparison with baseline models for 5-shot graph classification. Consistent with the results in the node classification task, SDGCL exhibits better accuracy across various label rates, with the performance gaps between SDGCL and baseline models being more pronounced for lower label rates. These results underscore the significance of SDGCL in scenarios where labeled data is scarce.

**Impact of Data Augmentation**. We also study the influence of graph augmentation on the few-shot graph classification task. As shown in Figure 5, the combination of three augmentation strategies yields the best performance, highlighting the significance of generating enough contrastive pairs during model training.

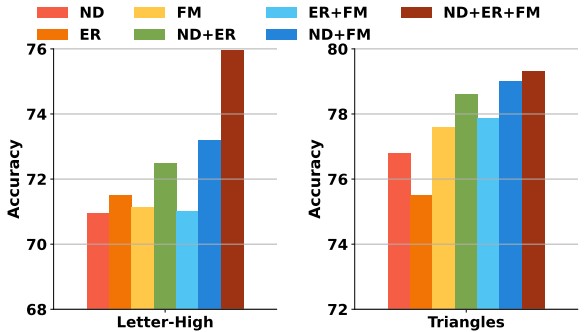

Figure 5: Impact of data aug. on graph classification.

Table 4: Info. loss for graph classification.

| Method | Layer | Letter-High | Triangles |
|--------|-------|-------------|-----------|
| GSM-GAT | GNN-1 | 1286.67 | 1520.94 |
|  | GNN-2 | 1250.54 | 1487.42 |
|  | FC | 1105.96 | 1364.11 |
| ASMAML | GNN-1 | 1023.23 | 1402.09 |
|  | GNN-2 | 901.60 | 1221.45 |
|  | GNN-3 | 899.12 | 1102.12 |
| SDGCL-I | GNN-1 | 932.23 | 1276.77 |
|  | GNN-2 | 920.58 | 1219.91 |
|  | FC-3 | 892.03 | 1107.56 |
| SDGCL-T | GNN-1 | **792.65** | **923.35** |
|  | GNN-2 | **780.59** | **894.09** |
|  | FC | **765.73** | **885.35** |

**Loss Information Comparison**. We also compare the amount of discarded graph information between our proposed model and a selected baseline method (GSM-GAT), as shown in Table 4. Notably, our model (SDGCL) discards less graph information than the baseline, which suggests that SDGCL can effectively capture and leverage more relevant information for graph classification tasks. This finding supports the superiority of our model in learning informative graph embeddings.

## 6    Conclusion

In this paper, we proposed a novel framework called SDGCL (**S**elf-**D**istilled **G**raph **F**ew-shot **L**earning) to address the limitations of existing Graph Few-shot Learning (GFL) models. Our framework is designed to learn generalized graph representations and overcome constraints in task-specific design. SDGCL leverages a self-distilled contrastive learning approach to enhance GFL by pre-training the GNN encoder with contrastive learning and then further improving it with knowledge distillation in a self-supervised manner. We also introduced an information-based method to compute the amount of discarded graph information by the GFL model. Our extensive experiments on multiple graph datasets demonstrated that SDGCL outperforms state-of-the-art baseline methods for both node classification and graph classification tasks in the few-shot scenario. The discarded information value further validates the superiority of SDGCL in learning node and graph embeddings.

## Acknowledgement

This work is partially supported by the NSF under grants CMMI-2146076 and Brandeis University. Any opinions, findings, conclusions or recommendations expressed in this material are those of the authors and do not necessarily reflect the views of any funding agencies.

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

# A    Dataset Details

We evaluate various models for few-shot node classification using five diverse datasets, including ogbn-arxiv (Hu et al., 2020a), Tissue-PPI (Hamilton et al., 2017), Fold-PPI (Zitnik & Leskovec, 2017), Cora (Sen et al., 2008), and Citeseer (Sen et al., 2008). These datasets range from citation networks to biochemical graphs. We provide detailed information on these datasets in Table 5.

Table 5: Statistics of datasets used in the node classification task.

| Dataset | # Graph | # Node | # Edge | # Feat. | # Label |
|---|---|---|---|---|---|
| ogbn-arxiv | 1 | 169,343 | 1,166,243 | 128 | 40 |
| Tissue-PPI | 24 | 51,194 | 1,350,412 | 50 | 10 |
| Fold-PPI | 144 | 274,606 | 3,666,563 | 512 | 29 |
| Cora | 1 | 2,708 | 10,556 | 1,433 | 7 |
| Citeseer | 1 | 3,327 | 9,228 | 3,703 | 6 |

For few-shot graph classification, we employ four distinct datasets (Chauhan et al., 2020): Reddit-12K, ENZYMES, Letter-High, and TRIANGLES, to conduct extensive empirical evaluations of various models. These datasets exhibit variations in terms of average graph size, ranging from small (e.g., Letter-High) to large (e.g., Reddit-12K). The statistics of datasets are reported in Table 6.

Table 6: Statistics of datasets used in the graph classification task.

| Dataset | Class # | | Graph # | | |
|---|---|---|---|---|---|
| | Train | Test | Training | Validation | Test |
| Letter-High | 11 | 4 | 1,330 | 320 | 600 |
| Triangles | 7 | 3 | 1,126 | 271 | 603 |
| Reddit-12K | 7 | 4 | 566 | 141 | 404 |
| Enzymes | 4 | 2 | 320 | 80 | 200 |

# B    Baseline Method Details

## B.1    Node Classification

**Graph embedding models:**

*node2vec (Grover & Leskovec, 2016)*: We use node2vec to generate node embeddings, then employ a FC layer as a predictor to classify nodes. We use the code at this link.[1]

*DeepWalk (Perozzi et al., 2014)*: Similar to node2vec, we use DeepWalk to generate node embeddings, then employ an FC layer as a predictor to classify nodes. We use the code at this link.[2]

**GNN-based models:**

*Meta-GNN (Zhou et al., 2019):* It combines MAML and simple graph convolution (SGC) to learn node embeddings. We use the code at this link.[3]

*FS-GIN (Xu et al., 2019):* This method uses GIN to learn node embeddings and only uses few-shot nodes to propagate loss and enable training. We use the code for GIN backbone at this link.[4]

*FS-SGC (Wu et al., 2019):* This model is similar to FS-GIN while changing GIN to SGC as GNN backbone. We use the code of SGC at this link.[5]

---

[1]https://shorturl.at/sEINW
[2]https://github.com/phanein/deepwalk
[3]https://github.com/ChengtaiCao/Meta-GNN
[4]https://github.com/weihua916/powerful-gnns
[5]https://github.com/Tiiiger/SGC

*No-Finetune (Huang & Zitnik, 2020):* This method trains a GCN on the support set and uses the trained backbone to classify samples in the meta-testing set. We use the code of GCN at this link.[6]

*Finetune (Triantafillou et al., 2019):* This method trains GCN on the meta-training set, and the model is fine-tuned on the meta-testing set. We use the code of GCN at this link.[6]

*KNN (Triantafillou et al., 2019):* This method trains a GNN on meta-training set. Then, it uses the label of the K-closest examples in the support set for each query example. We use the related code at this link.[7]

*ProtoNet (Triantafillou et al., 2019):* This method applies prototypical network on node embeddings processed by a neural network, which is trained under the standard meta-learning setting. We use the related code at this link.[8]

*MAML (Finn et al., 2017):* It is similar to ProtoNet but changes meta-learner from ProtoNet to MAML. We use the code at this link.[9]

*G-Meta (Huang & Zitnik, 2020):* This is a strong baseline for few-shot node classification. It uses GCN as GNN backbone to learn node embeddings based on local subgraphs. It further combines prototypical loss and MAML for model training. We use the code at this link.[9]

*TENT (Wang et al., 2022):* This is also a state-of-the-art baseline for few-shot node classification. It proposes task-adaptive node classification framework to make node-level, class-level, and task-level adaptations. We utilize the code at this link.[10]

### B.2   Graph Classification

**Graph embedding models:**

*WL (Shervashidze et al., 2011):* It uses KNN search on the output embeddings of WL. We use the code of WL at this link.[11]

*Graphlet (Shervashidze et al., 2009):* It uses Graphlet Kernel to decompose a graph and generates graph embeddings. We use the code of Graphlet at this link.[12]

*AWE (Ivanov & Burnaev, 2018):* It uses KNN search on the output embeddings of AWE. We use the code of AWE at this link.[12]

*Graph2Vec (Narayanan et al., 2017):* This method applies KNN search on the output embeddings of Graph2Vec. We use the code of Graph2Vec at this link.[12]

**GNN-based models:**

*Diffpool (Lee et al., 2019):* It uses Diffpool with supervised loss to generate graph embeddings. We use the code of Diffpool at this link.[13]

*CapsGNN (Xinyi & Chen, 2018):* This method applies CapsGNN to generate graph embeddings with supervised training. We use the code of CapsGNN backbone at this link.[14]

*GIN (Xu et al., 2019):* This model applies GIN to generate graph embeddings with supervised training. We use the code of GIN backbone at this link.[4]

*GIN-KNN (Xu et al., 2019):* Similarly, this model implements GIN to generate graph embeddings while it switches the MLP classifier to the KNN algorithm. We use the code of GIN backbone at this link.[4]

---

[6]https://shorturl.at/lwFPR
[7]https://shorturl.at/etBO8
[8]https://shorturl.at/erQU9
[9]https://github.com/mims-harvard/G-Meta
[10]https://github.com/SongW-SW/TENT
[11]https://github.com/BorgwardtLab/P-WL
[12]https://github.com/paulmorio/geo2dr
[13]https://github.com/RexYing/diffpool
[14]https://github.com/benedekrozemberczki/CapsGNN

*GSM-GCN (Chauhan et al., 2020):* This is a strong model (with GCN as backbone) for few-shot graph classification. We follow the default settings in the original paper and use the code at this link.[15]

*GSM-GAT (Chauhan et al., 2020):* This is a strong model (with GAT as backbone) for few-shot graph classification. We follow the default settings in the original paper and use the code at this link.[15]

*AS-MAML (Ma et al., 2020):* It is a state-of-the-art model for few-shot graph classification. We follow the default settings in the original paper and use the code at this link.[16]

## C  Additional Experiment Results

The following section presents supplementary experiment outcomes on three extra datasets for few-shot node classification (Fold-PPI, Cora, and Citeseer) and two additional datasets for few-shot graph classification (Reddit-12K and Enzymes).

### C.1  Few-Shot Node Classification Results

**Impact of Shot Number**. As shown in Figure 6, we present our model's performance on the other three datasets (Fold-PPI, Cora, and Citeseer) under different shot numbers (1 to 5) compared with selected baseline methods. It is evident that SDGCL outperforms baseline models consistently across various shot numbers, demonstrating its effectiveness and generalization ability.

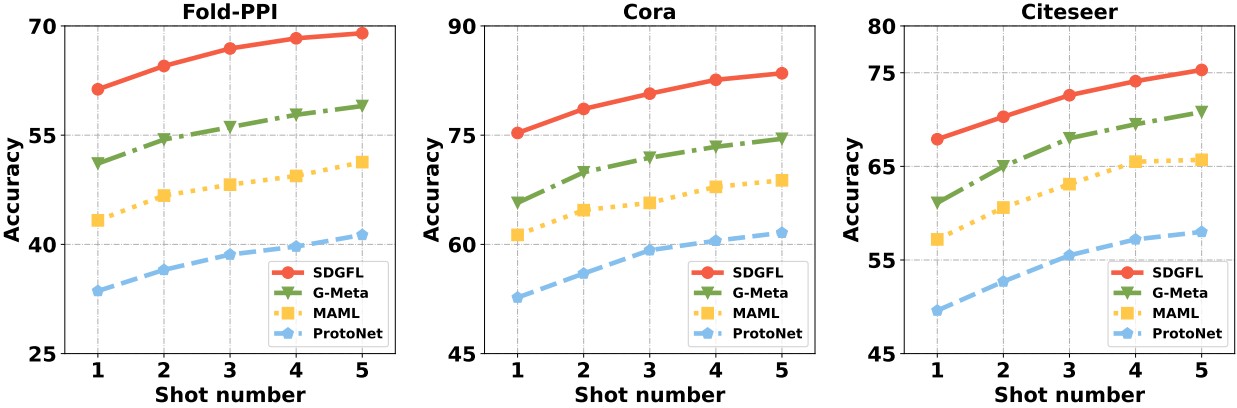

Figure 6: Impact of shot number on node classification.

**Impact of Training Label Rate**. Figure 7 displays the performance of our model under different training label rates compared to baseline models for 3-shot node classification. It is evident that our model achieves better accuracy across different label rates.

**Impact of Data Augmentation**. Figure 8 presents the impact of graph augmentation on node classification. It can be observed that the combination of three augmentation strategies yields the best performance.

**Discarded Information Comparison**. The discarded graph information of different models are shown in Table 7. Obviously, the amount of information discarded by SDGCL is less than that by the baseline models.

### C.2  Few-Shot Graph Classification Results

**Impact of Shot Number**. Figure 9 shows the performance of SDGCL under different shot numbers (5, 10, 15, 20) compared to some selected baselines. From this figure, it is evident that our model consistently outperforms baseline methods across different shot numbers.

---

[15]https://github.com/chauhanjatin10/GraphsFewShot
[16]https://github.com/NingMa-AI/AS-MAML

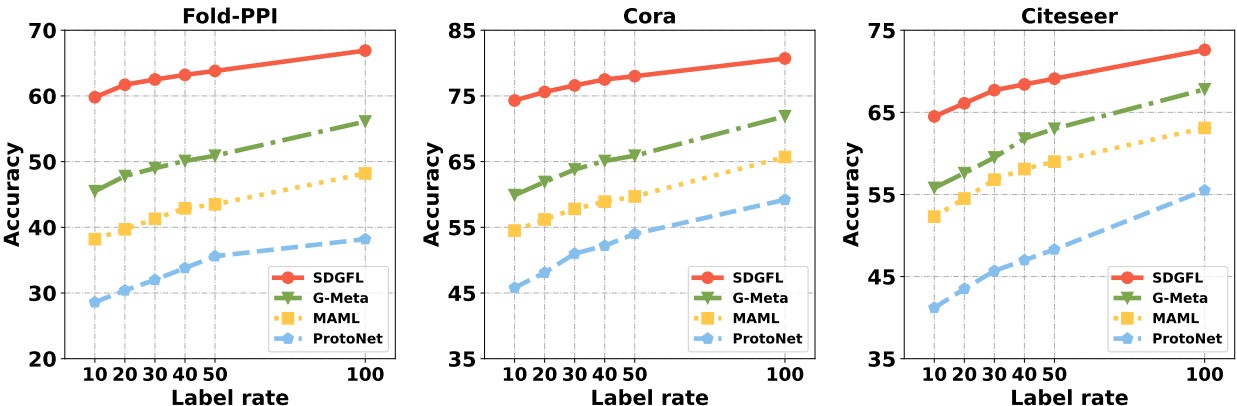

Figure 7: Impact of training label rate on node classification.

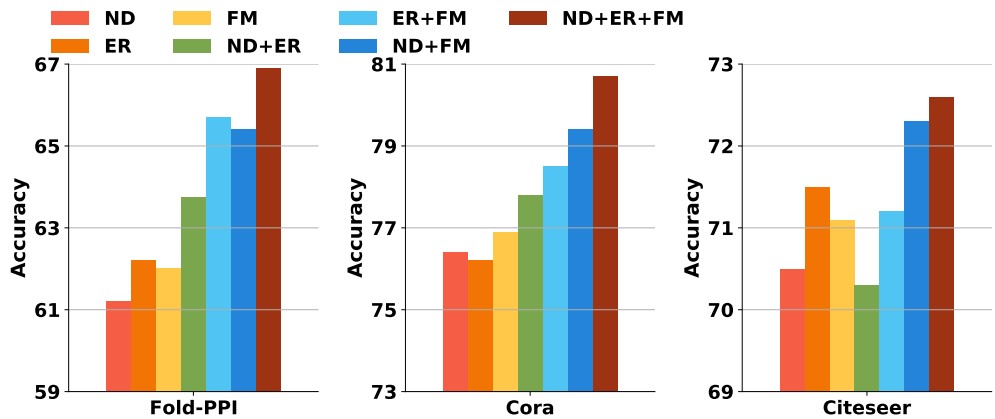

Figure 8: Impact of data augmentation on node classification.

**Impact of Training Label Rate**. Figure 10 presents the performance of SDGCL under different training label rates for 5-shot graph classification, compared to baseline models. As can be seen from the figure, SDGCL achieves better performance across different label rates.

Table 7: Discarded information for node classification.

| Method | Layer | Discarded Information | | |
| --- | --- | --- | --- | --- |
| | | Fold-PPI | Cora | Citeseer |
| Meta-GNN | GNN-1 | 618.52 | 361.70 | 134.20 |
| | GNN-2 | 612.11 | 357.47 | 126.75 |
| | FC | 591.13 | 327.49 | 116.55 |
| G-Meta | GNN-1 | 585.31 | 356.57 | 129.04 |
| | GNN-2 | 592.38 | 355.63 | 119.72 |
| | FC | 553.25 | 318.87 | 100.10 |
| SDGCL-I | GNN-1 | 509.10 | 340.72 | 111.90 |
| | GNN-2 | 511.10 | 326.35 | 105.01 |
| | FC | 461.52 | 316.21 | 92.95 |
| SDGCL-T | GNN-1 | **426.23** | **332.39** | **104.04** |
| | GNN-2 | **418.90** | **314.65** | **88.33** |
| | FC | **384.23** | **306.98** | **78.25** |

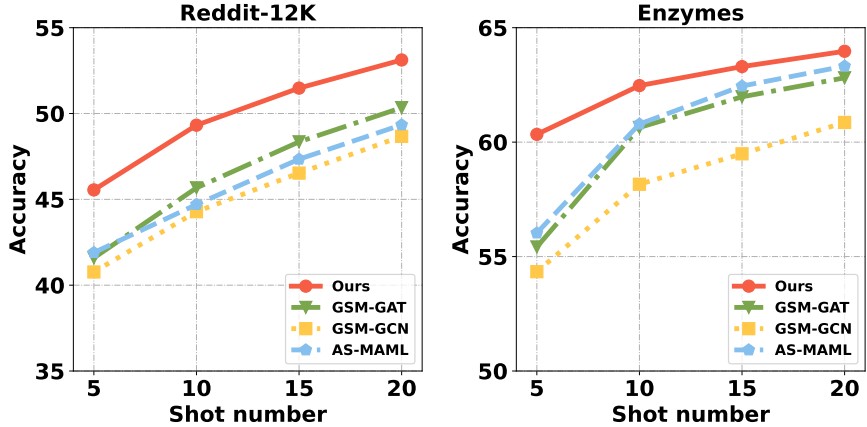

Figure 9: Impact of shot number on graph classification.

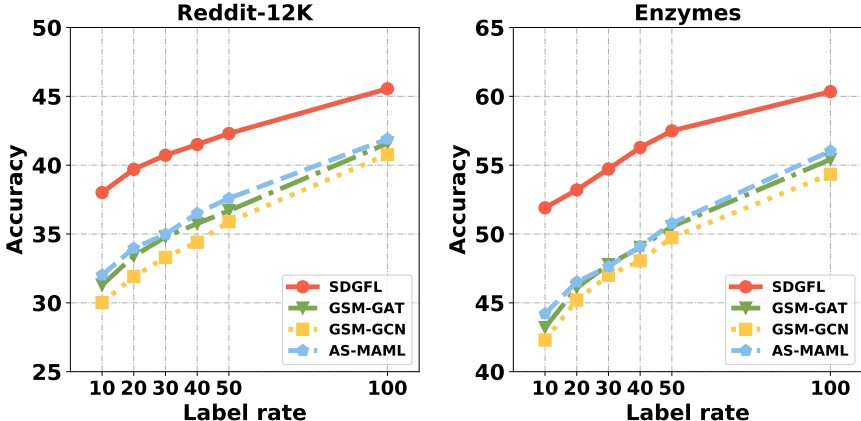

Figure 10: Impact of training label rate on graph classification.

**Impact of Data Augmentation**: The effect of graph augmentation on graph classification is demonstrated in Figure 11. We observe that combining the three augmentation strategies results in the best performance.

**Discarded Information Comparison**: We present the discarded graph information of various methods in Table 8. The table shows that SDGCL discards less information compared to the baseline models.

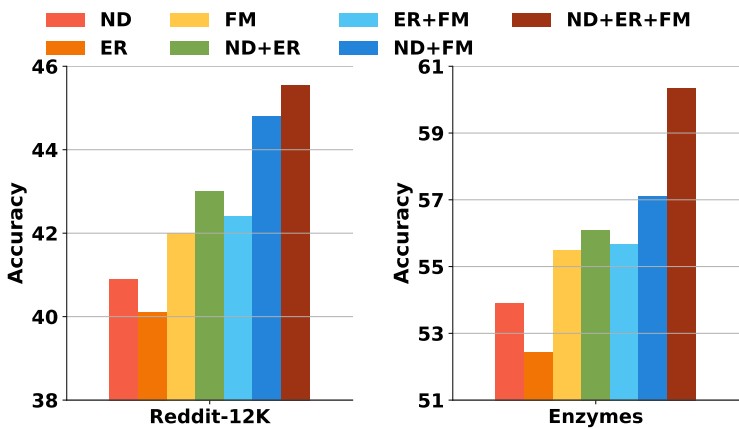

Figure 11: Impact of data augmentation on graph classification.

Table 8: Discarded information for graph classification.

| Method | Layer | Discarded Information | |
|---|---|---|---|
| | | Reddit-12K | Enzymes |
| AS-MAML | GNN-1 | 5455.46 | 2455.54 |
| | GNN-2 | 5024.15 | 2243.25 |
| | FC | 4937.99 | 2125.91 |
| GSM-GAT | GNN-1 | 5401.30 | 2273.80 |
| | GNN-2 | 5183.87 | 2128.04 |
| | FC | 4936.76 | 1958.18 |
| SDGCL-I | GNN-1 | 5323.04 | 1864.98 |
| | GNN-2 | 5098.54 | 1788.97 |
| | FC | 4802.03 | 1759.87 |
| SDGCL-T | GNN-1 | **4823.91** | **1787.98** |
| | GNN-2 | **4546.23** | **1698.35** |
| | FC | **4329.39** | **1585.33** |

