# OpenReview forum: "Mind the Gap: Mitigating the Distribution Gap in Graph Few-shot Learning"
_TMLR — Accepted by TMLR_

### Review · Reviewer_Gaj3 · 2023-04-05

**Summary Of Contributions:**

This paper deals with the Graph Few-shot Learning (GFL) problem with the help of self-supervised contrastive learning.

Specifically, self-supervised contrastive graph learning makes use of existing contrastive learning good practices (Grill et al., 2020; Furlanello et al., 2018; You et al., 2020). The process consists of two stages: pure self-supervised learning and self-distillation graph contrastive learning.
After self-supervised learning, general graph few-shot learning techniques are used for the problem utilizing MAML for the graph few-shot learning phase.
Moreover, a Quantitative Measurement of GFL using Mutual information is proposed to evaluate the layerwise information for the learned representation model.

The technique contributions lie in (1) designing and varying the effectiveness of self-distillation contrastive learning in the Graph Few-shot Learning scenario, (2) quantitative measurement of GFL utilizing Mutual information.

Experimentally, the proposed method is conducted on two problem settings: node classification and graph classification. Some of the experiments verify the effectiveness.


**Audience:**

Yes

**Claims And Evidence:**

Yes

**Requested Changes:**

Please see the above experimental weaknesses and typos.

**Strengths And Weaknesses:**

## Strengths
- The practical design and combination of self-distillation contrastive learning strategies seem to be effective for Graph Few-shot learning.
- A new measurement is proposed to evaluate graph few-shot representation.
- The experiments are generally comprehensive and somehow verify the effectiveness of the proposed method for two settings and the proposed measurement.
- The paper is easy to follow.

## Weaknesses
- Since this paper re-designs/modifies existing contrastive learning techniques which have already been verified effective in image classification/graph learning. The technical contribution is not quite significant. But overall, it is also good to understand contrastive learning especially self-distillation behaviors in a few-shot graph context.
- My main concern is the unfair and insufficient comparisons in the experimental section.
> - In Table 1 and Table 2, the proposed method has both Inductive and Transductive settings, but all baselines are not marked 'I' or 'T.' Since the Transductive setting has been proven to improve the few-shot performance (Liu et al. ICLR 2019), this is unfair without indicating the I or T of the baselines.
> - In Figure 2, TENT is not compared, which is the best baseline method of Table 1.
> - In Table 2, TENT is not compared.
> - In Table 4, the strong baseline ASMAML is not compared.
- Typo or format issues.
> - The second paragraph of Related work starts from a new line, while the first paragraph does not. It is better to be consistent.
> - ``generative or contrastive, according to their learning tasks (Liu et al., 2020; Sohn et al., 2020).."
> - In Table 1, ``The best results are highlighted in bold while the best baseline results are underlined.". This is not the case. Please fix in Table 1.

---

### Review · Reviewer_D1oC · 2023-04-09

**Summary Of Contributions:**

This paper proposes an SDGCL framework that overcomes the negative impacts of the distribution shift. SDGCL is task and data-independent, which can be used for general graph mining purposes. An information-based method is also proposed to evaluate the quality of learned embeddings of the model. This paper conducts extensive experiments to demonstrate the effectiveness of the paper.

**Audience:**

Yes

**Claims And Evidence:**

Yes

**Requested Changes:**

1.	The paper should give a more explanation of the mentioned weakness.

**Strengths And Weaknesses:**

Strength:
1. The target issues of the paper are meaningful and worth exploring. The motivation is clear. This submission gives a valuable implementation of such an idea and presents good results.
2. The proposed method has been validated on different datasets and tasks.
Weakness:
1.	The proposed method is not novel, and a similar self-supervised graph learning method has already been explored in Infograph[A] and GCA[B]. This paper only uses the distillation method for improving the GFL.
2.	The paper lacks experiments in DBLP and Amazon-E. The compared method TENT in Table 1 has been evaluated in these datasets. More comparison with TENT is needed.
3.	In Table 1, TENT performs better in both 3-shot and 5-shot settings, more explanation is needed.

[A] Fan-Yun Sun, Jordan Hoffmann, Vikas Verma, and Jian Tang. Infograph: Unsupervised and semi-supervised graph-level representation learning via mutual information maximization. In ICLR, 2020.
[B] Yanqiao Zhu, Yichen Xu, Feng Yu, Qiang Liu, Shu Wu, and Liang Wang. Graph Contrastive Learning with Adaptive Augmentation. In WWW, 2021.

---

### Review · Reviewer_tbQa · 2023-04-14

**Summary Of Contributions:**

This paper proposes a novel framework called Self-Distilled Graph Few-shot Learning (SDGFL) to address the issue of label scarcity in deep graph learning models. SDGFL uses a self-distilled contrastive learning procedure to pre-train a graph encoder with unlabeled data, which is then used to distill a student model with a contrastive loss. This distilled model is fed to GFL to enhance generalization ability and mitigate the distribution gap. The proposed framework is task and data-independent, making it a versatile tool for general graph mining purposes. The effectiveness of SDGFL is evaluated using an information-based measurement, and comprehensive experiments show that it outperforms state-of-the-art baselines on various graph mining tasks in the few-shot scenario.

**Audience:**

Yes

**Broader Impact Concerns:**

I think this work on pretraining + meta-learning for graph few-shot learning in general has an impact on the GNN community.
I do have concerns about the novelty of the method (both contrastive/distillation and MAML part), but I suppose it is a bit hard to improve in a short period of time.

**Claims And Evidence:**

No

**Requested Changes:**

To be accepted, I think the bottom line is to provide a convincing rationale and motivation about why using the self-distillation step.

**Strengths And Weaknesses:**

# Strengths

* The authors did comprehensive experiments and ablation studies. I appreciate that.
* Paper is well-written and well-organized.

# Weaknesses
1. I completely don’t understand the purpose of the self-distillation step.
2. Why not use the teacher model for the meta-training phase? I know authors studied teacher-only in Tables 1 and 3, but why does the self-distillation step can enhance the pre-training, and in what sense? I carefully check the paper and I cannot find lines that specifically explain the reason. If I missed anything please correct me.
3. What does this line mean? “Both pre-training and the distillation steps can operate at the meta-training and meta-testing phases without requiring labeled data.” I thought the pre-training/self-distillation phase and the meta-learning/testing phase are completely separated?
4. I think the MAML-based Graph Few-Shot Learning is mainly a plug-and-play study and is not novel.
5. The "stop grad" part in contrastive learning is not new. The authors should at least properly cite and give credit to [1].

[1] "Bootstrap Your Own Latent A New Approach to Self-Supervised Learning" 2020

---

### Review · Reviewer_Ywqz · 2023-04-24

**Summary Of Contributions:**

The paper proposes a general procedure for training graph neural networks for few-shot node/graph classification. The first stage is constrastive pretraining, followed by the model-agnostic meta learning as the second stage. Empirical imporvements over some baselines are observed.

**Audience:**

Yes

**Claims And Evidence:**

No

**Requested Changes:**

See the weakness section.

**Strengths And Weaknesses:**

**Strengths**:

- The paper is generally well writte, clearly structured and quite easy to follow.
- The empirical results seem good on most of the performed tasks.


**Weaknesses**:

- The paper seems to be a simple ad-hoc combintation of contrastive learning and meta-learning on graphs. Most importantly, why these two methods are suited for each other is not well explained and justified.
- I think the roles of contrastive pretraining and meta-learning are somehow contraditive. Both methods are trying to learn some kind of initialization (i.e. an initial model). Why they are combined togethor needs more justifications.
- The distillation stage also seems questionable. I don't see why we actually need this. Could you elaborate on this?
- Many important ablation studies are missing. Is every single component (contrastive pretraining, distillation and meta-learning) in this pipeline really necessary? It is not convincing if there is no ablation on these components.
- Since the novelty is not in specific components of this pipeline, why this pipeline can improve performance requires deeper understanding. It will strengthen the paper if the authors can provide a theoretical justification or a comprehensive empirical study.